# Combining Statistical Evidence When Evidence Is Measured by Relative Belief

**DOI:** 10.3390/e27060654

**Published:** 2025-06-18

**Authors:** Michael Evans

**Affiliations:** Department of Statistical Sciences, University of Toronto, Toronto, ON M5G 1Z5, Canada; mevansthree.evans@utoronto.ca

**Keywords:** combining priors, statistical evidence, preserving consensus, Jeffrey conditionalization, ancillarity

## Abstract

The problem of combining statistical evidence concerning an unknown, contained in each of the *k* Bayesian inference bases, is discussed. This can be considered as being related to the problem of pooling *k* priors to determine a consensus prior, but the focus here is instead on combining a measure of statistical evidence to obtain a consensus measure of statistical evidence. The linear opinion pool is seen to have the most appropriate properties for this role. In particular, linear pooling preserves a consensus with respect to the evidence, and other rules do not. While linear pooling does not preserve prior independence, it is shown that it still behaves appropriately with respect to the expression of statistical evidence in such a context. For the more general problem of combining statistical evidence, where the priors as well as the sampling models may differ, Jeffrey conditionalization plays a key role.

## 1. Introduction

Suppose that *k* different experts choose models and priors for a statistical analysis concerning a common quantity of interest Ψ which is a parameter or a future value. A problem then arises as to how the resulting statistical analyses should be combined so that the inferences presented can serve as a consensus inference. If all the models are the same, then this is the well-known problem of combining priors and this is covered by our discussion here. Even for the problem of combining priors, however, a somewhat different point-of-view is taken. A particular measure of statistical evidence is adopted, as discussed in Section 3, such that the data set, sampling model and prior leads to either evidence in favor of or against each possible value of Ψ. Throughout the paper, the word ‘evidence’ is often used alone, but it always refers to the statistical evidence rather than some alternative kind of evidence. In this paper, it is concluded that the linear pooling rule, see [1], is the most appropriate for combining evidence.

The purpose then is to determine a consensus on what the evidence indicates by combining the measures of statistical evidence rather than focusing on combining priors. Since the primary goal of a statistical analysis is to express what the evidence says about Ψ, this seems appropriate. Also, it is perfectly reasonable that some analyses express evidence against while others express evidence in favor but the combined expression of the evidence is one way or the other, see Section 2.

Before discussing the combination approach, however, it is necessary to be more precise about the problem and distinguish between somewhat different contexts where the problem can arise. It will be supposed here that Ψ is a parameter of interest but prediction problems are easily handled by a slight modification, see Example 3. Let $M=\{f_{\theta }:\theta \in \Theta \}$ denote a generic statistical model and ψ=Ψ(θ), where Ψ:Θ→Ψ is onto and, to save notation, the function and its range have the same symbol.

**Context I.** Suppose there is a single statistical model M for the data *x* and *k* distinct priors $\pi_{i}$ so there are *k* inference bases $I_{i}=\left(x,M,\pi_{i}\right)$ for i=1,…,k. It is assumed that the conditional priors $\pi_{i}\left(\cdot \;|\;\psi \right)$ on the nuisance parameters are all the same, as is satisfied when Ψ(θ)=θ. This situation arises when there is a group of analysts who agree on M and perhaps use a default prior for the nuisance parameters, while each member puts forward a prior for Ψ.**Context II.** Suppose there are *k* data sets, models, and priors as given by the inference bases $I_{i}=\left(x_{i},M_{i},\pi_{i}\right)$ for i∈{1,…,k} and there is a common characteristic of interest $\psi =\Psi (\theta_{i})$ with the true value of ψ being the same for each model, as will occur when ψ corresponds to some real-world quantity. Strictly speaking, the function Ψ also depends on *i* when the parameter spaces $\Theta_{i}$ differ, but we suppress this dependence because each context is referring to the same real-world object.

It is a necessary part of any statistical analysis that a model be checked to see whether or not it is contradicted by the data, namely, determining if it is the case that the data lies in the tails of each distribution in the model. So in any situation where there is a lack of model fit, it is necessary to modify that component of the inference base. Similarly, each prior needs to be checked for prior–data conflict, namely, is there an indication that the true value lies in the tails of the prior, see [2,3]. If such a conflict is found, then the prior needs to be modified, see [4]. For the purpose of the discussion here, however, it is assumed that all the models and priors have passed such checks. A salutary effect of a lack of prior–data conflict, is that it rules out the possibility of trying to combine priors which have little overlap in terms of where they place their mass.

Given an inference base I=(x,M,π) and interest in ψ=Ψ(θ), a Bayesian analysis has an important consistency property. In particular, this inference base is equivalent, for inference about ψ, to the inference base $I=(x,M_{\Psi },\pi_{\Psi })$ where $\pi_{\Psi }$ is the marginal prior on ψ and $M_{\Psi }=\left\{m_{\psi }:\psi \in \Psi \right\}$ with$$m_{\psi }\left(x\right)=E_{\pi (\cdot \;|\;\psi )}\left(f_{\theta }\left(x\right)\right),$$
the prior predictive density of the data obtained by integrating out the nuisance parameters via the conditional prior π(·|ψ) for θ given ψ=Ψ(θ). So, for example, the posterior $\pi_{\Psi }\left(\cdot \;|\;x\right)$ for ψ obtained via these two inference bases is the same and moreover the evidence about ψ is also the same. This result has implications for the combination strategy as it is really the inference bases $I_{i}=\left(x,M_{\Psi },\pi_{i\Psi }\right)$ that are relevant in Context I and it is the inference bases $I_{i}=\left(x_{i},M_{i\Psi },\pi_{i\Psi }\right)$ that are relevant in Context II, namely, nuisance parameters are always integrated out before combining.

Note that if, in a collection of inference bases $I_{i}=\left(x_{i},M_{i},\pi_{i}\right)$ for i∈{1,…,k}, all the models are based on sampling from the same basic model, and the conditional priors $\pi_{i}\left(\cdot \;|\;\psi \right)$ on the nuisance parameters are all he same, then it makes sense to combine the data sets to $x=(x_{1},\ldots ,x_{k})$ with the combined model M being based on the sample *x* so we are in Context I as only the marginal priors $\pi_{\Psi }$ differ. This combination would not be possible if the conditional priors $\pi_{i}\left(\cdot \;|\;\psi \right)$ on the nuisance parameters differed as then the models $M_{i\Psi }$ will be different. We will assume hereafter that the following principle has been applied.

**Combining inference bases rule**: all data sets that are assumed to arise from the same set of basic distributions are combined whenever the conditional priors on the nuisance parameters are the same, so that separate data sets are associated with truly distinct models and/or priors.This rule ensures that any combination reflects true differences among the beliefs concerning where the truth about Ψ lies as there is agreement on the other ingredients. It is assumed hereafter that this is applied before the inference bases $I_{i}$ are determined. Note that, even if the basic model is the same for each i, when the conditional priors on the nuisance parameters differ, then this is Context II.

In Section 2 a general family of rules for combining priors with given weights is presented. In Section 3 the problem of combining evidence for Context I is analyzed, with given weights for the respective priors, and the linear pooling combination rule is seen to have most appropriate properties with respect to evidence. In Section 3.1 the problem of determining appropriate weights is considered. In Section 4 the problem for Context II is discussed and a proposal is made for a rule that generalizes the rule for Context I. The rule for Context I possesses a natural consistency property as the combined evidence is the same whether considered as a mixture of the evidence arising from each inference base or obtained directly from the combined prior and the corresponding posterior. In particular, it is Bayesian in this generalized sense which differs from being externally Bayesian as discussed in [5]; see Section 3. This is not the case for Context II, however, because of differing nuisance parameters and ambiguities in the definition of the likelihood, but Jeffrey conditionalization provides a meaningful interpretation, at least when all the inference bases contain the same data.

The problem of combining priors has an extensive literature. Ref. [6] is a basic reference and reviews can be found in [7,8,9,10]. Ref. [11] is a significant recent application. Broadly speaking there are mathematical approaches and behavioral approaches. The mathematical approach provides a formal rule, as in Section 2, while the behavioral approach provides methodology for a group of proposers to work towards a consensus through mutual interaction. For example, ref. [12] considers the elicitation procedure where quantities concerning the object of interest are elicited by each member of a group and then the average elicited values are used to choose the prior. Ref. [13] adopts a supra-Bayesian approach where the data generated during the elicitation process is conditioned on in a formal Bayesian analysis to choose a prior in a family on which an initial prior has been placed. Ref. [14] presents an iterative methodology for a group of proposers to work towards a consensus prior based upon each proposer seeing how far their proposal deviated from a current grouped proposal. While the behavioral approach has a number of attractive features, there are also reservations as indicated by Kahneman in [15].

The focus in this paper is on presenting a consensus assessment of the evidence via a combination of the evidence that each analyst obtains. In particular, the priors $\pi_{i}$ need not arise via the same elicitation procedure and the proposers may not be aware of other proposals although the approach does not rule this out. Also, utility functions, necessary for decisions, are not part of the development as these may indeed lead to conflicts with what the evidence indicates and they are not generally checkable against the data as with models and priors. The assessment of statistical evidence as the primary driver of statistical methodology is a theme that many authors have pursued, for example, see [16,17,18,19]. Ref. [20] reviews many of the attempts to provide a precise definition of the concept of statistical evidence. Ref. [21] discusses the importance of the amalgamation of evidence, although evidence there references a more general concept than what is considered here.

Throughout the paper the densities of probability distributions will be represented by lower case symbols and the associated probability measure will be represented by the same symbol in upper case. For example, if a prior density is denoted by π, then the prior probability measure will be denoted by Π with the posterior density denoted π(·|x) and the posterior probability measure by Π(·|x).

## 2. Combining Priors with Given Prior Weights

Let $\alpha =\left(\alpha_{1},\ldots ,\alpha_{k}\right)\in S_{k}$ the (k−1)-dimensional simplex for some k≥2 and, for now, suppose that α is given. While general combination rules could be considered, attention is restricted here to the power means of densities$$\pi_{t,\alpha }=\left\{\begin{array}{cc} c_{t}\left(\alpha ,\pi_{\cdot }\right){\left\{\sum_{i=1}^{k}\alpha_{i}\pi_{i}^{t}\right\}}^{1/t}, & t\neq 0,\pm \infty \\ c_{0}\left(\alpha ,\pi_{\cdot }\right)\exp\left\{\sum_{i=1}^{k}\alpha_{i}\log\pi_{i}\right\}, & t=0\\ c_{-\infty }\left(\alpha ,\pi_{\cdot }\right)\min\left\{\pi_{1},\ldots ,\pi_{k}\right\}, & t=-\infty \\ c_{\infty }\left(\alpha ,\pi_{\cdot }\right)\max\left\{\pi_{1},\ldots ,\pi_{k}\right\}, & t=\infty \end{array}\right.$$
where $\pi_{\cdot }=\left(\pi_{1},\ldots ,\pi_{k}\right)$ and, for any α and sequence of nonnegative functions $g_{\cdot }=\left(g_{1},\ldots ,g_{k}\right)$ defined on Θ, then $c_{t}\left(\alpha ,g_{\cdot }\right)$ is the relevant normalizing constant. Note that $\pi_{-\infty ,\alpha }$ and $\pi_{\infty ,\alpha }$ do not depend on α.

For each θ the mean ${\left\{\sum_{i=1}^{k}\alpha_{i}\pi_{i}^{t}\left(\theta \right)\right\}}^{1/t}$ is nondecreasing in t, see [22], and two of the means are equal everywhere iff all priors are the same. Since $c_{1}\left(\alpha ,\pi_{\cdot }\right)=1,$ this implies that $c_{t}\left(\alpha ,\pi_{\cdot }\right)$ is finite for all α whenever t≤1. If t>1 is to be considered, then it is necessary to check on the integrability of the mean so that a proper prior is obtained and this will be assumed to hold whenever the case t>1 is referenced. When Θ is finite, this is not an issue.

The following result characterizes how the posterior behaves in terms of a combination of the individual posteriors. Let $m_{i}$ denote the *i*-th prior predictive density based on prior $\pi_{i},$$m_{t,\alpha }$ denote the prior predictive density obtained using the $\pi_{t,\alpha }$ prior and • denotes component-wise multiplication of two vectors of the same dimension.

**Proposition 1.** 
*For Context I, the posterior based on $\pi_{t,\alpha }$ equals*

$$\pi_{t,\alpha }\left(\theta \;|\;x\right)=\left\{\begin{array}{cc} c_{t}\left(\alpha ,m_{\cdot }\left(x\right)•\pi_{\cdot }\left(\cdot \;|\;x\right)\right){\left\{\sum_{i=1}^{k}\alpha_{i}m_{i}^{t}\left(x\right)\pi_{i}^{t}\left(\theta \;|\;x\right)\right\}}^{1/t}, & t\neq 0\\ c_{0}\left(\alpha ,\pi_{\cdot }\left(\cdot \;|\;x\right)\right)\exp\left\{\sum_{i=1}^{k}\alpha_{i}\log\pi_{i}\left(\theta \;|\;x\right)\right\}, & t=0\\ c_{-\infty }\left(\alpha ,m_{\cdot }\left(x\right)•\pi_{\cdot }\left(\cdot \;|\;x\right)\right)\min_{i=1\ldots ,k}m_{i}\left(x\right)\pi_{i}\left(\theta \;|\;x\right), & t=-\infty \\ c_{\infty }\left(\alpha ,m_{\cdot }\left(x\right)•\pi_{\cdot }\left(\cdot \;|\;x\right)\right)\max_{i=1\ldots ,k}m_{i}\left(x\right)\pi_{i}\left(\theta \;|\;x\right), & t=\infty \end{array}\right.$$

*and $m_{t,\alpha }\left(x\right)/c_{t}\left(\alpha ,\pi \right)\leq \left(\geq \right)m_{1,\alpha }\left(x\right)$ when t≤(≥)1.*


**Proof.** The expressions for $\pi_{t,\alpha }\left(\cdot \;|\;x\right)$ for t≠0 are obvious and$$\begin{array}{cc} \; & \pi_{0,\alpha }\left(\theta \;|\;x\right)=c_{0}\left(\alpha ,m_{\cdot }\left(x\right)•\pi_{\cdot }\left(\cdot \;|\;x\right)\right)\exp\left\{\sum_{i=1}^{k}\alpha_{i}\log m_{i}\left(x\right)\pi_{i}\left(\theta \;|\;x\right)\right\}\\ \; & ={\left\{\int_{\Theta }\prod_{i=1}^{k}m_{i}^{\alpha_{i}}\left(x\right)\prod_{i=1}^{k}\pi_{i}^{\alpha_{i}}\left(\theta \;|\;x\right)\;d\theta \right\}}^{-1}\prod_{i=1}^{k}m_{i}^{\alpha_{i}}\left(x\right)\pi_{i}^{\alpha_{i}}\left(\theta \;|\;x\right) \end{array}$$
so the factor $\prod_{i=1}^{k}m_{i}^{\alpha_{i}}\left(x\right)$ cancels giving the result. Finally,$$m_{t,\alpha }\left(x\right)=c_{t}\left(\alpha ,\pi \right)\int_{\Theta }{\left\{\sum_{i=1}^{k}\alpha_{i}m_{i}^{t}\left(x\right)\pi_{i}^{t}\left(\theta \;|\;x\right)\right\}}^{1/t}\;d\theta$$
and this is bounded above (below) by$$c_{t}\left(\alpha ,\pi \right)\int_{\Theta }\left\{\sum_{i=1}^{k}\alpha_{i}m_{i}\left(x\right)\pi_{i}\left(\theta \;|\;x\right)\right\}\;d\theta =c_{t}\left(\alpha ,\pi \right)m_{1,\alpha }\left(x\right)$$
when t≤(≥)1 which gives the inequality. □

So the posterior is always proportional to a power mean of the individual posteriors of the same degree as the power mean of the priors but, excepting the t=0 case, the weights have changed and when t=−∞ or t=−∞ the prior and posterior do not depend on α. The posterior resulting when t=1 is(1)$$\pi_{1,\alpha }\left(\theta \;|\;x\right)=\sum_{i=1}^{k}\left(\frac{\alpha_{i}m_{i}\left(x\right)}{\sum_{i=1}^{k}\alpha_{i}m_{i}\left(x\right)}\right)\pi_{i}\left(\theta \;|\;x\right)=\sum_{i=1}^{k}\frac{\alpha_{i}m_{i}\left(x\right)}{m_{1,\alpha }\left(x\right)}\pi_{i}\left(\theta \;|\;x\right),$$
and so is a linear combination of the individual posteriors but with different weights than the prior. The case t=1 is called the *linear opinion pool*, see [1], and when t=0 it is called the *logarithmic opinion pool*.

The weights staying constant from a priori to a posteriori property for $\pi_{0,\alpha },$ or even independence from the weights, may seem like an appealing property but, as discussed in Section 3, these combination rules have properties that make them inappropriate for combining evidence. A combination rule is said to be *externally Bayesian* when the rule for combining the posteriors is the same as the rule for combining the priors. As shown in [5,23], logarithmic pooling is characterized by being externally Bayesian while linear pooling only satisfies this when there is a dictatorship, namely, $\alpha_{i}=1$ for some i, as otherwise the weights differ. Proposition 2 (iii) shows, however, that there is a sense in which linear pooling can be considered as Bayesian.

Linear pooling has a number of appealing properties.

**Proposition 2.** 
*For Context I, linear pooling satisfies the following:*

*(i) the prior probability measures satisfies the same combination rule as the densities, namely, $\Pi_{1,\alpha }=\sum_{i=1}^{k}\alpha_{i}\Pi_{i}$ and similarly for the posterior measures,*

*(ii) marginal priors obtained from $\Pi_{1,\alpha }$ are equal to the same combination of the marginal priors obtained from the $\Pi_{i}$, and this is effectively the only rule with this property among all possible combination rules,*

*(iii) if (i,θ,x) is given joint prior distribution with density $\alpha_{i}\pi_{i}\left(\theta \right)f_{\theta }\left(x\right)$, then the posterior density of θ is given by (Equation 1) and the weight $\alpha_{i}m_{i}\left(x\right)/m_{1,\alpha }\left(x\right)$ is the posterior probability of the index i.*



**Proof.** The proof of (i) is obvious while (ii) is proved in [24] and holds here with no further conditions. For (iii), note that $\pi_{i}$ is the conditional prior of θ given *i* and $f_{\theta }$ is the conditional density of *x* given θ. Once *x* is observed, the posterior of (i,θ) is then given by $\alpha_{i}\pi_{i}\left(\theta \right)f_{\theta }\left(x\right)/m_{1,\alpha }\left(x\right)$ which implies that the marginal posterior of θ is (Equation 1) and the posterior probability of *i* is $\alpha_{i}m_{i}\left(x\right)/m_{1,\alpha }\left(x\right)$. □

The significance of (i) is that the other combination rules considered here do not exhibit such simplicity and require more computation to obtain the measures. Property (ii) implies that integrating out nuisance parameters before or after combining does not affect inferences about a marginal parameter ψ in Context I, as conditional priors on the nuisance parameters being the same, implies that the marginal models for ψ are all the same. Ref. [25] proves a similar result allowing for negative $\alpha_{i}$. Property (iii) shows that both the prior $\pi_{1,\alpha }$ and the posterior $\pi_{1,\alpha }\left(\cdot \;|\;x\right)$ arise via valid probability calculations when α is known. A possible interpretation of this is that $\alpha_{i}$ represents the combiner’s prior belief in how well the *i*-th prior represents appropriate beliefs concerning the true value of θ relative to the other priors. The posterior weight $\alpha_{i}m_{i}\left(x\right)/m_{1,\alpha }\left(x\right)$ is then the appropriate modified belief after seeing the data, as the factor $m_{i}\left(x\right)/m_{1,\alpha }\left(x\right)$ reflects how well the *i*-th inference base has done at predicting the observed data relative to the other inference bases. This is a somewhat different interpretation than that taken by [26] where $\alpha_{i}$ represents the combiner’s prior belief that the *i*-th inference base is the true one which, in this context, does not really apply.

One commonly cited negative property of linear pooling, see [27], is that if *A* and *C* are independent events for each $\Pi_{i},$ then generally $\Pi_{1,\alpha }\left(A\cap C\right)\neq \Pi_{1,\alpha }\left(A\right)\Pi_{1,\alpha }\left(C\right).$ It is to be noted that if also one of $\Pi_{i}\left(A\right)$ or $\Pi_{i\alpha }\left(C\right)$ is constant in i, then independence is preserved and this will be seen to play a role in linear pooling behaving appropriately when considering statistical evidence, see Proposition 4(ii) and the discussion thereafter.

## 3. Combining Measures of Evidence in Context I

The criterion for choosing an appropriate combination should depend on how statistical evidence is characterized, as using the evidence to determine inferences is the ultimate purpose of a statistical analysis. The underlying idea concerning evidence used here is the following principle.

**Principle of Evidence:** there is evidence in favor of the value ψ if $\pi_{\Psi }\left(\psi \;|\;x\right)>\pi_{\Psi }\left(\psi \right),$ there is evidence against the value ψ if $\pi_{\Psi }\left(\psi \;|\;x\right)<\pi_{\Psi }\left(\psi \right),$ and no evidence either way if $\pi_{\Psi }\left(\psi \;|\;x\right)=\pi_{\Psi }\left(\psi \right)$.

The basic idea is that, if the data has lead to an increase in the belief that ψ is the true value from a priori to a posteriori, then the data contains evidence in favor of of ψ, etc. This interpretation is obviously the case when the prior is a discrete distribution and it also holds in the continuous case via a limit argument, see [17]. The principle of evidence does not require that a specific numerical measure of evidence be chosen only that any measure used be consistent with this principle, namely, that there is a cut-off such that the numerical value greater than (less than) the cut-off corresponds to evidence in favor of (against) as indicated by the principle. The relative belief ratio$$RB_{\Psi }\left(\psi \;|\;x\right)=\frac{\pi_{\Psi }\left(\psi \;|\;x\right)}{\pi_{\Psi }\left(\psi \right)},$$
the ratio of the posterior to the prior, with the cut-off 1, is used here as it has a number of good properties, see [17]. It is also particularly appropriate for the combination of the evidence as easily interpretable formulas result. The Bayes factor is also a valid measure of evidence, but there are many reason to prefer the relative belief ratio to measure evidence as discussed in [28].

The next result examines the behavior of the combination rules of Section 2 with respect to evidence and is stated initially for the full model parameter θ in Context I. For this $RB_{i}\left(\theta \;|\;x\right)$ is the relative belief ratio for θ that results from the *i*-th inference base $I_{i}=\left(x_{i},M_{i},\pi_{i}\right)$ and $RB_{t,\alpha }\left(\theta \;|\;x\right)$ is the relative belief ratio for θ that results from combining the *k* priors using the *t*-th power mean combination rule.

**Proposition 3.** 
*For Context I, the relative belief ratio for θ based on the $\pi_{t,\alpha }$ prior satsifies*

(2)
$$RB_{t,\alpha }\left(\theta \;|\;x\right)=\frac{m_{1,\alpha }\left(x\right)}{m_{t,\alpha }\left(x\right)}RB_{1,\alpha }\left(\theta \;|\;x\right).$$



**Proof.** Using $RB_{i}\left(\theta \;|\;x\right)=f_{\theta }\left(x\right)/m_{i}\left(x\right)$ and $f_{\theta }\left(x\right)=\sum_{i=1}^{k}\alpha_{i}f_{\theta }\left(x\right),$ then$$\begin{array}{cc} RB_{t,\alpha }\left(\theta \;|\;x\right) & =\frac{f_{\theta }\left(x\right)}{m_{t,\alpha }\left(x\right)}=\frac{m_{1,\alpha }\left(x\right)}{m_{t,\alpha }\left(x\right)}\sum_{i=1}^{k}\frac{\alpha_{i}m_{i}\left(x\right)}{m_{1,\alpha }\left(x\right)}RB_{i}\left(\theta \;|\;x\right)=\frac{m_{1,\alpha }\left(x\right)}{m_{t,\alpha }\left(x\right)}RB_{1,\alpha }\left(\theta \;|\;x\right). \end{array}$$□

This result shows the value of using the relative belief ratio to express evidence as the combination rule, at least for power means, is quite simple and natural. Notice too that if there are only *l* distinct priors, then the combination rules for the priors, posteriors and relative belief ratios are really only based on these distinct priors and the weights change only by summing the $\alpha_{i}$ that correspond to common priors.

The result in Proposition 3 is another indication that the correct way to combine priors, from the point of view of measuring evidence, is via linear pooling as $RB_{t,\alpha }\left(\theta \;|\;x\right)$ is always proportional to $RB_{1,\alpha }\left(\theta \;|\;x\right).$ The constant multiplying $RB_{1,\alpha }\left(\theta \;|\;x\right)$ in (Equation 2) suggests that finding *t* that minimizes $m_{1,\alpha }\left(x\right)/m_{t,\alpha }\left(x\right)$, leads to the power mean prior that maximizes the amount of mass the prior places at $\theta_{true},$ see Proposition 5 (iv). But there is a significant reason for preferring $RB_{1,\alpha }\left(\theta \;|\;x\right)$ over the other possibilities. Suppose that $RB_{i}\left(\theta \;|\;x\right)<1$ for all i or $RB_{i}\left(\theta \;|\;x\right)>1$ for all i. Then it is clear that $RB_{1,\alpha }\left(\theta \;|\;x\right)<1$ in the first case and $RB_{1,\alpha }\left(\theta \;|\;x\right)>1$ in the second case. In the first case there is a consensus that there is evidence against θ being the true value and in the second case there is a consensus that there is evidence in favor of θ being the true value. In other words $RB_{1,\alpha }$ is consensus preserving and this seems like a necessary property for any approach to combining evidence.

A formal definition is now provided which takes into account that sometimes $RB_{i}\left(\theta \;|\;x\right)=1$, indicating that there is no evidence either way, which implies that the *i*-th inference base is agnostic about whether or not θ is the true value.

**Definition** A rule for combining evidence about a parameter is called *consensus preserving* if, whenever at least one of the inference bases indicates evidence in favor of (against) a value of the parameter and the remaining inference bases do not give evidence against (in favor), then the rule gives evidence in favor of (against) the value and if no inference base indicates evidence one way or the other, then neither does the combination.

The following property is immediately obtained for linear pooling.

**Proposition 4.** 
*For Context I, whenever $\alpha_{i}>0$ for all i, then (i) $RB_{1,\alpha }$ is consensus preserving and (ii) whenever $RB_{i}\left(\theta \;|\;x\right)\leq \left(\geq \right)1$ for all i, then $RB_{1,\alpha }\left(\theta \;|\;x\right)=1$ iff $RB_{i}\left(\theta \;|\;x\right)=1$ for all i.*


The property of preserving consensus is similar to the unanimity principle for priors, see [7], which says that if all the priors are the same, then the combination rule must give back that prior and all the power mean rules satisfy this.

Proposition 4 (ii) indicates that linear pooling deals correctly with independent events at least with respect to evidence. For note that, for probability measure *P* and events *A* and *C* satisfying P(A∩C)>0, then *A* and *C* are statistically independent iff RB(A|C)=P(A∩C)/P(A)P(C)=1. So, independence is equivalent to saying that the occurrence of *C* provides no evidence concerning the truth or falsity of *A* and conversely. Now consider the statistical context and suppose $RB_{i}\left(\theta \;|\;x\right)=1$ and further suppose that all the probabilities are discrete. This implies that $f_{\theta }\left(x\right)=m_{i}\left(x\right)$ which implies that the joint prior density at (θ,x) factors as $f_{\theta }\left(x\right)\pi_{i}\left(\theta \right)=m_{i}\left(x\right)\pi_{i}\left(\theta \right)$ and so the events {θ} and {x} are statistically independent in the *i*-th inference base. If this holds for each i, then $m_{i}\left(x\right)$ is constant in *i* and so indeed $RB_{1,\alpha }\left(\theta \;|\;x\right)=1$ implies that these events are independent when the prior is the linear pool. With a continuous prior, then $RB_{i}\left(\theta \;|\;x\right)=1$ can also happen, but typically this event has prior probability 0.

It is of interest to determine whether or not any of the other rules based on the means are consensus preserving. The inequality in Proposition 1 and Proposition 3 imply that, when t≤1, then $RB_{t,\alpha }\left(\theta \;|\;x\right)\geq RB_{1,\alpha }\left(\theta \;|\;x\right)/c_{t}\left(\alpha ,\pi \right)$ with $c_{t}\left(\alpha ,\pi \right)\geq 1,$ with the inequality typically strict when t<1. This suggests that $RB_{t,\alpha }$ might even contradict the consensus of evidence in favor. A similar argument holds for t>1. The following example shows that generally the combination rules based on power means of priors are not consensus-preserving.

**Example 1.** 
*Power means of priors are not generally consensus preserving.*

*Suppose $X=\left\{0,1\right\},\Theta =\left\{a,b\right\},f_{a}\left(0\right)=1/4,f_{b}\left(0\right)=2/3$ and x=0 is observed. There are two priors given by $\pi_{1}\left(a\right)=p_{1}$ and $\pi_{2}\left(a\right)=p_{2}.$ Then*

$$\begin{array}{cc} m_{1}\left(0\right) & =\left(8-5p_{1}\right)/12,\;RB_{1}\left(a\;|\;0\right)=3/\left(8-5p_{1}\right),\\ m_{2}\left(0\right) & =\left(8-5p_{2}\right)/12,\;RB_{2}\left(a\;|\;0\right)=3/\left(8-5p_{2}\right), \end{array}$$

*so both inference bases give evidence against when $p_{1}<1,p_{2}<1.$ When $p_{i}=1$, then $RB_{i}\left(a\;|\;0\right)=1$ so no evidence either way is obtained from the data when a statistician is categorical in their beliefs. Note being categorical in your beliefs is a possible choice provided it does not lead to prior–data conflict. In this case, there is no prior–data conflict even with $p_{1}=1$ since there is a reasonable probability of observing x=0 when a=1.*

*When α=(1/2,1/2), so the two priors are being given equal weight, then*

$$\begin{array}{cc} \pi_{1,\alpha }\left(a\right) & =(p_{1}+p_{2})/2,\\ m_{1,1/2}\left(0\right) & =\left(m_{1}\left(0\right)+m_{2}\left(0\right)\right)/2=\left(8-p_{1}-p_{2}\right)/24,\\ RB_{1,\alpha }\left(a\;|\;0\right) & =\left(m_{1}\left(0\right)RB_{1}\left(a\;|\;0\right)+m_{2}\left(0\right)RB_{2}\left(a\;|\;0\right)\right)/2m_{1,1/2}\left(0\right)\\ \; & =6/(8-p_{1}-p_{2}). \end{array}$$

*When $p_{1}=1,p_{2}=1/2,$ so statistician 1 is categorical in their beliefs, and α=(1/2,1/2), then*

$$\begin{array}{cc} RB_{1}\left(a\;|\;0\right) & =1.00,\;\pi_{1}\left(a\;|\;0\right)=1.00,\\ RB_{2}\left(a\;|\;0\right) & =0.55,\;\pi_{2}\left(a\;|\;0\right)=0.27,\\ RB_{1,\alpha }\left(a\;|\;0\right) & =0.71,\;\pi_{1,\alpha }\left(a\;|\;0\right)=0.53. \end{array}$$


*So, statistician 1 finds no evidence either way for a being the true value from the data and this is because, when a prior is categorical, the data is irrelevant as it does not change beliefs. Statistician 2 finds evidence against a and the posterior probability of 0.27 indicates reasonably strong belief in a not being the true value. Linear pooling indicates evidence against a, as it should, and the posterior probability of 0.53 indicates weak belief in a not being the true value and this decrease in the strength of the evidence against is because of the first statistician’s complete confidence in the truth of a and the combination of beliefs. Note that α=(1/2,1/2) indicates complete indifference between the quality of the statisticians priors but, if we put less weight on the first statistician’s prior, then the evidence against and its strength moves closer to that of statistician 2.*


*Now consider logarithmic pooling where $\pi_{0,\alpha }\left(a\right)=p_{1}^{\alpha }p_{2}^{1-\alpha }/\left(p_{1}^{\alpha }p_{2}^{1-\alpha }+{\left(1-p_{1}\right)}^{1-\alpha }{\left(1-p_{2}\right)}^{\alpha }\right).$ In particular, with $p_{1}=1,$ then $\pi_{0,\alpha }\left(a\right)=1$, no matter what α is, and $m_{0,\alpha }\left(0\right)=f_{a}\left(0\right)$ for every α. By Proposition 3, with $p_{1}=1,$*

$$RB_{0,\alpha }\left(a\;|\;0\right)=\frac{m_{1,\alpha }\left(0\right)}{m_{0,\alpha }\left(0\right)}RB_{1,\alpha }\left(a\;|\;0\right)=1,\;\pi_{0,\alpha }\left(a\;|\;0\right)=1,$$

*which indicates no evidence for or against a being the true value. Therefore, logarithmic pooling is not consensus preserving. The illogicality of this is readily apparent as it suggests that no evidence has been found one way or the other and that is not the case.*

*Next consider the case t=−∞, so $\pi_{-\infty ,\alpha }\left(a\right)=\min\left\{p_{1},p_{2}\right\}/\left(\min\left\{p_{1},p_{2}\right\}+\min\left\{1-p_{1},1-p_{2}\right\}\right).$ When $p_{1}=1$ and $p_{2}<1,$ then $\pi_{-\infty ,\alpha }\left(a\right)=p_{2}/\left(1-p_{1}+p_{2}\right)=1$ and $RB_{-\infty ,\alpha }\left(a\;|\;0\right)=1$ which shows that this combination rule is also not consensus preserving. In this context, and based on numerical computation, it seems that $RB_{t,\alpha }\left(a\;|\;0\right)=1$ for every t≤0 and so all of these combination rules are not consensus preserving and note that this includes the harmonic mean combination rule. If there is evidence against (in favor of) an event, then a property of the relative belief ratio gives that there is evidence in favor of (against) its complement and, if there is no evidence either way for an event, then there is no evidence either way for its complement, see [17], Proposition 4.2.3 (i). So in this example the priors $\pi_{0,\alpha }$ and $\pi_{-\infty ,\alpha }$ also do not preserve consensus with respect to θ=b.*


So far no case has been found where a combination based on a power mean actually reverses a consensus and it is a reasonable conjecture, based on many examples, that this will never happen but a proof is not obvious. Also, other power means may preserve consensus but currently we do not have such a result. Logarithmic pooling could be considered as the main rival to linear pooling, but Example 1 shows that it does not preserve consensus.

There is another interesting consequence of Proposition 3 which is relevant when the goal is to estimate θ. The natural estimate is the relative belief estimate $\theta \left(x\right)=\arg\sup_{\theta }RB\left(\theta \;|\;x\right)$ where the accuracy of θ(x) is assessed by the *plausible region* Pl(x)={θ:RB(θ|x)>1}, the set of values for which there is evidence in favor. For example, the “size” of Pl(x) and its posterior content together provide an a posteriori measure of how accurate θ(x) is. Ideally we want Pl(x) “small” and its posterior content high. The size of Pl(x) can be measured in various ways such as Euclidean volume, cardinality, or prior content, with the context determining which is most suitable. Note that it is easy to show in general that RB(θ(x)|x)>1 so θ(x)∈Pl(x) provided RB(θ|x) is not 1 for all θ, which only occurs when the data indicates nothing about the true value.

**Corollary 1.** 
*Whenever $RB_{t,\alpha }\left(\theta \;|\;x\right)$ is not 1 for all θ and $\alpha_{i}>0$ for all i, then $\arg\sup_{\theta }RB_{t,\alpha }\left(\theta \;|\;x\right)=\arg\sup_{\theta }RB_{1,\alpha }\left(\theta \;|\;x\right).$*


So the estimate of θ based on maximizing the evidence in favor is determined by linear pooling for every *t*. It is not the case, however, that the plausible region is independent of *t* because of the constant $m_{1,\alpha }\left(x\right)/m_{t,\alpha }\left(x\right).$

The following underscores the role of linear pooling in preserving consensus.

**Corollary 2.** 
*The set $\{\theta :RB_{i}\left(\theta \;|\;x\right)>1$ for all $i\}=\cap_{i=1}^{k}Pl_{i}\left(x\right)\subset Pl_{1,\alpha }\left(x\right)$ and $\min_{i}\Pi_{i}\left(Pl_{i}\left(x\right)\;|\;x\right)\leq \Pi_{1,\alpha }\left(Pl_{1,\alpha }\left(x\right)\;|\;x\right)\leq \max_{i}\Pi_{i}\left(Pl_{i}\left(x\right)\;|\;x\right).$*


So the set of θ where there is a consensus that there is evidence in favor is always contained in the plausible region determined by linear pooling. A similar comment applies to the *implausible region* which is the set of all values where there is evidence against. While it might be tempting to quote the region $\cap_{i=1}^{k}Pl_{i}\left(x\right),$ there is no guarantee that any of the relative belief estimates will be in this set, whether determined by $RB_{1,\alpha }\left(\cdot \;|\;x\right)$ or any of the $RB_{i}\left(\cdot \;|\;x\right).$

The situation with respect to the assessment of the hypothesis $H_{0}:\theta =\theta_{0}$ is a bit different. Clearly, if $RB_{i}\left(\theta_{0}\;|\;x\right)>\left(<\right)1$ for all i, so there is a consensus that there is evidence in favor of (against) $H_{0},$ then $RB_{1,\alpha }\left(\theta_{0}\;|\;x\right)$ preserves this consensus. In general, when the evidence in favor of or against $H_{0}$ is assessed via a relative belief ratio $RB(\theta_{0}\;|\;x),$ then the posterior probability $\Pi (RB\left(\theta_{0}\;|\;x\right)\leq RB\left(\theta_{0}\;|\;x\right)\;|\;x)$ can be taken as a measure of the strength of the evidence, see [17]. In the context under discussion here, it follows from (Equation 2) that the event $\left\{\theta :RB_{t,\alpha }\left(\theta \;|\;x\right)\leq RB_{t,\alpha }\left(\theta_{0}\;|\;x\right)\right\}=\left\{\theta :RB_{1,\alpha }\left(\theta \;|\;x\right)\leq RB_{1,\alpha }\left(\theta_{0}\;|\;x\right)\right\}$ for all t. Of course, the posterior probability of this event will depend on *t* but linear pooling completely determines the event.

Now suppose that interest is in the quantity ψ=Ψ(θ) and the assumptions of Context I hold so that prior beliefs only differ concerning the value of ψ, which implies that the inference bases only differ with respect to the priors on ψ. This situation may arise when the analysts all agree to use a common default prior on the nuisance parameters. Then we can treat ψ as the model parameter for the common model $\left\{m_{\psi }:\psi \in \Psi \right\}$ and the relevant linear pooling rule is(3)$$RB_{1,\alpha ,\Psi }\left(\psi \;|\;x\right)=\sum_{i=1}^{k}\frac{\alpha_{i}m_{i}\left(x\right)}{m_{1,\alpha }\left(x\right)}RB_{i,\Psi }\left(\psi \;|\;x\right),$$
where $RB_{i,\Psi }\left(\psi \;|\;x\right)$ is the relative belief ratio for ψ obtained from the *i*-th inference base. Note that the results derived for θ also apply for inferences about ψ. In general it can be expected that some inference bases will indicate evidence in favor of ψ being the true value and some will indicate evidence against, but $RB_{1,\alpha ,\Psi }\left(\psi \;|\;x\right)$ will indicate evidence one way or the other or even perhaps no evidence either way. This depends on the values assumed by the $RB_{i,\Psi }\left(\psi \;|\;x\right)$ as well as the weights $\alpha_{i}m_{i}\left(x\right)/m_{1,\alpha }\left(x\right)$, with larger values of a weight leading to a greater contribution to the overall inferences by the corresponding inference base. This aspect is discussed in Section 3.1.

Consider now the context where ${\overset{˘}{x}}_{n}=\left(x_{1},\ldots ,x_{n}\right)$ is an i.i.d. sample. The following result gives the consistency of this approach when the model parameter space Θ is finite. Such results will hold more generally but require some mathematical constraints on densities and this is not pursued further here. Let $\psi_{1,\alpha }\left({\overset{˘}{x}}_{n}\right)=\arg\max_{\psi }RB_{\Psi ,1,\alpha }\left(\psi \;|\;{\overset{˘}{x}}_{n}\right)$ be the relative belief estimate of ψ based on linear pooling. All the convergence results are almost everywhere as n→∞ with the proofs in the Appendix A.

**Proposition 5.** 
*For Context I, suppose ${\overset{˘}{x}}_{n}=\left(x_{1},\ldots ,x_{n}\right)$ is an i.i.d. sample from a distribution in a model having a finite parameter space
Θ. and each prior for θ is everywhere positive on Θ. Then*

*(i) $RB_{1,\alpha ,\Psi }\left(\psi_{0}\;|\;{\overset{˘}{x}}_{n}\right)\to I_{\left\{\psi_{true}\right\}}\left(\psi_{0}\right)/\pi_{1,\alpha ,\Psi }\left(\psi_{0}\right)$ and*

$$\Pi_{1,\alpha ,\Psi }\left(RB_{1,\alpha ,\Psi }\left(\psi \;|\;{\overset{˘}{x}}_{n}\right)\leq RB_{1,\alpha ,\Psi }\left(\psi_{0}\;|\;{\overset{˘}{x}}_{n}\right)\right)\;\left|\;{\overset{˘}{x}}_{n}\right)\to I_{\left\{\psi_{true}\right\}}\left(\psi_{0}\right),$$


*(ii) $\psi_{1,\alpha }\left({\overset{˘}{x}}_{n}\right)\to \psi_{true},Pl_{1,\alpha ,\Psi }\left({\overset{˘}{x}}_{n}\right)\to \left\{\psi_{true}\right\}$ and $\Pi_{1,\alpha ,\Psi }\left(Pl_{1,\alpha ,\Psi }\left({\overset{˘}{x}}_{n}\right)\;|\;{\overset{˘}{x}}_{n}\right)\to 1,$*

*(iii) $\alpha_{i}m_{i}\left(x\right)/m_{1,\alpha }\left(x\right)\to \alpha_{i}\pi_{i}\left(\theta_{true}\right)/\pi_{1,\alpha }\left(\theta_{true}\right),$*

*(iv) $m_{1,\alpha }\left(x\right)/m_{t,\alpha }\left(x\right)\to \pi_{1,\alpha }\left(\theta_{true}\right)/\pi_{t,\alpha }\left(\theta_{true}\right).$*



Noting that when $1/\pi_{1,\alpha ,\Psi }\left(\psi_{0}\right)>1,$ then Proposition 5 (i) says that the evidence in favor of (against) $H_{0}:\Psi \left(\theta \right)=\psi_{0},$ based on the combination, goes to categorical when $H_{0}$ is true (false). Part (ii) says that the relative belief estimate based on the combination is consistent. Part (iii) implies that, when the priors are equally weighted, then the inference base whose prior gives the largest value to the true value will inevitably have the largest weight in determining the combined evidence. As previously mentioned, part (iv) suggests choosing *t* to minimize the ratio $m_{1,\alpha }\left(x\right)/m_{t,\alpha }\left(x\right)$ as this can be associated with choosing the power combination prior that maximizes the amount of belief the prior places on the true value. This has the unnatural consequence, however, that the prior is being determined by the data.

Our overall conclusion, based on the results established here, is that linear pooling is the most natural way to combine evidence among the power means. As such, attention is restricted to this case hereafter. Various authors, when discussing the combination of priors, have come to a similar conclusion. For example, ref. [10], when considering the full spectrum of methods for combining priors, contains the following assertion, “In general, it seems that a simple, equally weighted, linear opinion pool is hard to beat in practice”. The results developed here support such a conclusion when considering evidence.

### 3.1. Determining the Prior Weights

The discussion so far has assumed that α is known but arguments or methodologies for choosing α need to be considered. There are several possible approaches to determining a suitable choice of the prior weights and nothing novel is proposed here. As previously mentioned, the $\alpha_{i}$ can represent the combiner’s beliefs concerning how well the *i*-th prior represents appropriate beliefs about θ. The combiner’s beliefs should of course be based upon experience or knowledge concerning the various proposers of the priors. In absence of such knowledge then uniform weights, namely, α=(1/k,…,1/k), seem reasonable. Ref. [29] provides a good survey of various approaches to choosing α. Also, ref. [30,31] present a novel iterative approach to determining a consensus α among the proposers.

In Context I notice that the weights $\alpha_{i}m_{i}\left(x\right)/m_{1,\alpha }\left(x\right)$ only depend on the data through some function of the value of the minimal sufficient statistic (mss) for the model. So, for example, if the priors are distinct and equally weighted via α=(1/k,…,1/k), then the weight of the *i*-th prior is $m_{i}\left(x\right)/\left(m_{1}\left(x\right)+\ldots +m_{k}\left(x\right)\right)$ and so more weight is given to those inference bases that do a better job, relatively speaking, of predicting a priori the observed value of this function of the mss. Since it is only the observed value of the mss that is relevant for inference, this seems sensible. There is the possibility, however, to weight some priors more than others for a variety of reasons.

A prior can also be placed on α, the results examined for a number different choices of α and summarized in a way that addresses the issue of whether or not the inferences are sensitive to α. For example, suppose the goal is to determine if there is evidence for or against the hypothesis $H_{0}:\Psi \left(\theta \right)=\psi_{0}.$ For a given weighting $\alpha_{0},$ the evidence for or against will be determined by the value $RB_{1,\alpha_{0},\Psi }\left(\psi_{0}\;|\;x\right).$ Accordingly, a Dirichlet prior with mode at $\alpha_{0}$ and with some degree of concentration around this point could be used to assess the robustness of the combination inferences. In particular, for each generated value of α from the prior, one can record whether evidence in favor of or against $H_{0}$ was obtained together with the strength of the evidence. If a great proportion of the results gave results similar to those obtained with the weights $\alpha_{0},$ then this would provide some assurance that the conclusions drawn are robust to deviations. A similar approach can be taken to estimation problems where the relative belief estimate is given by $\psi \left(x\right)=\arg\sup_{\psi }RB_{1,\alpha_{0},\Psi }\left(\psi \;|\;x\right).$ When Ψ is 1-dimensional then a histogram of the estimates obtained in the simulation and histograms of the prior and posterior contents of $Pl_{\Psi }\left(x\right)$ will provide an indication of the dependence on $\alpha_{0}.$

## 4. The General Problem

The general Context II is more complicated and an overall solution is not proposed here. Rather, a special case is considered when there is a common data set x. So, *k* analysts are making inference about the same real-world object Ψ, based on the same data, but they are using possibly different models and different priors. Since Context II covers Context I, it is necessary that any rule proposed for such situations agrees with what is determined for Context I when that applies.

While it may seem reasonable to take the prior on ψ to be the linear mixture $\pi_{1,\alpha ,\Psi }=\sum_{i=1}^{k}\alpha_{i}\pi_{i,\Psi },$ this cannot be viewed as a marginal prior obtained by integrating out nuisance parameters from $\sum_{i=1}^{k}\alpha_{i}\pi_{i}$, as in Context I, because the nuisance parameters vary with i. Also, even if we elected to use this prior, the overall posterior does not have a clear definition as it is not obvious how to form the likelihood. As such, a different approach and justification is required.

The simplest approach to characterizing the evidence in Context II is to use(4)$$RB_{1,\alpha ,\Psi }^{*}\left(\psi \;|\;x\right)=\sum_{i=1}^{k}\frac{\alpha_{i}m_{i}\left(x\right)}{m_{1,\alpha }\left(x\right)}RB_{i,\Psi }\left(\psi \;|\;x\right),$$
where again $RB_{i,\Psi }\left(\psi \;|\;x\right)$ and $m_{i}\left(x\right)$ arise from the *i*-th inference base and $m_{1,\alpha }\left(x\right)=\sum_{i=1}^{k}\alpha_{i}m_{i}\left(x_{i}\right)$. This will agree with the answer obtained in Context I when it applies, but generally $RB_{1,\alpha ,\Psi }^{*}\left(\psi \;|\;x\right)$ is not the ratio of the posterior of ψ to its prior. As such, it cannot be claimed that (Equation 4) is a valid characterization of the evidence, as $RB_{1,\alpha ,\Psi }\left(\psi \;|\;x\right)$ is in Context I, even though each $RB_{i,\Psi }\left(\psi \;|\;x\right)$ is a valid measure of evidence.

One approach to defining a prior and a posterior in Context II is to use the argument known as Jeffrey conditionalization, see [32]. This involves considering the probabilities on the partition given by i∈{1,…,k} completely separately from the probabilities on ψ given i. If we knew i, then standard conditioning leads to $\pi_{i,\Psi }\left(\cdot \;|\;x\right)$ as the expression of posterior beliefs about ψ. But *i* is unknown and all that is available are the probabilities given by α and Jeffrey conditionalization suggests $\sum_{i=1}^{k}\alpha_{i}\pi_{i,\Psi },$ and $\sum_{i=1}^{k}\alpha_{i}\pi_{i,\Psi }\left(\cdot \;|\;x\right)$ as the appropriate expressions of prior and posterior beliefs.

But note that, based on the *k* inference bases, $\alpha_{i}\pi_{i}\left(\theta_{i}\right)f_{\theta_{i}}\left(x\right)$ can be thought of as the prior probability distribution for $\left(i,\theta_{i},x\right)$ which leads to $\alpha_{i}\pi_{i,\Psi }\left(\psi \right)m_{i}\left(x\;|\;\psi \right)$ as the prior for (i,ψ,x). Since the likelihood $m_{i}\left(x\;|\;\psi \right)$ depends on i, however, Context I does not apply. Still, the joint prior for (i,x) is $\alpha_{i}m_{i}\left(x\right)$ and, after observing x, from the combiner’s point-of-view, this leads to the posterior probability $\alpha_{i}m_{i}\left(x\right)/m_{1,\alpha }\left(x\right)$ for i. From the *i*-th analyst’s viewpoint, $\pi_{i,\Psi }\left(\cdot \;|\;x\right)$ gives the appropriate posterior for ψ and so, applying the Jeffrey conditionalization idea, leads to the combination posterior for ψ given by(5)$$\pi_{1,\alpha ,\Psi }^{*}\left(\psi \;|\;x\right)=\sum_{i=1}^{k}\frac{\alpha_{i}m_{i}\left(x\right)}{m_{1,\alpha }\left(x\right)}\pi_{i,\Psi }\left(\psi \;|\;x\right).$$
This could be considered as a generalization of Jeffrey’s idea as now the probabilities on the partition elements and ψ both depend on the data. Furthermore, extending Jeffrey’s idea to the combination of the measurement of evidence, we obtain (Equation 4). While this is not formally a valid measure of evidence, (Equation 4) will satisfy all the properties of linear pooling established for Context I with the exception of Proposition 2 (iii). In particular, $RB_{1,\alpha ,\Psi }^{*}\left(\psi \;|\;x\right)$ will preserve a consensus about evidence in favor or against. A key reason for not using $\pi_{1,\alpha ,\Psi }^{*}=\sum_{i=1}^{k}\alpha_{i}\pi_{i,\Psi }$ and $\pi_{1,\alpha ,\Psi }^{*}\left(\cdot \;|\;x\right)$ as the prior and posterior to determine the evidence, is that the nice properties of linear pooling are lost, see Example 3.

The following result characterizes what happens as sample size grows and is proved in the Appendix A. Again convergence is almost everywhere.

**Proposition 6.** 
*Suppose ${\overset{˘}{x}}_{n}=\left(x_{1},\ldots ,x_{n}\right)$ is an i.i.d. sample from a distribution in at least one of the models and each of the parameter spaces $\Theta_{i}$ is finite with the prior $\pi_{i}$ everywhere positive on $\Theta_{i}.$ Denoting the set of indices corresponding to the models containing the true distribution by J, then as n→∞*

*(i) $\alpha_{i}m_{i}\left({\overset{˘}{x}}_{n}\right)/m_{1,\alpha }\left({\overset{˘}{x}}_{n}\right)\to w_{i}=I_{J}\left(i\right)\alpha_{i}\pi_{i}\left(\theta_{itrue}\right)/\sum_{j\in J}\alpha_{j}\pi_{j}\left(\theta_{jtrue}\right)\geq 0$ and $\sum_{i=1}^{k}w_{i}=1\;$*

*(ii) $RB_{1,\alpha ,\Psi }^{*}\left(\psi \;|\;{\overset{˘}{x}}_{n}\right)\to I_{\left\{\psi_{true}\right\}}\left(\psi \right)\sum_{i=1}^{k}w_{i}/\pi_{i\Psi }\left(\psi \right)$ which is greater than 1 when $\psi =\psi_{true}$*

*(iii) $\lim\pi_{1,\alpha ,\Psi }^{*}\left(\psi \;|\;{\overset{˘}{x}}_{n}\right)=I_{\left\{\psi_{true}\right\}}\left(\psi \right).$*



So Proposition 6 shows that $RB_{1,\alpha ,\Psi }^{*}\left(\cdot \;|\;{\overset{˘}{x}}_{n}\right)$ and $\pi_{1,\alpha ,\Psi }^{*}\left(\cdot \;|\;{\overset{˘}{x}}_{n}\right)$ provide consistent inferences and the weights converge to appropriate values.

There is another significant difference between (Equation 4) and (Equation 3). In Context I the weights all depended on the data through the same function of a constant mss for the full common model. Furthermore, if A(x) is an ancillary statistic for the full model, then it is seen that the *i*-th weight satisfies $\alpha_{i}m_{i}\left(x\right)/m_{1,\alpha }\left(x\right)=\alpha_{i}m_{i}\left(x\;|\;A\left(x\right)\right)/m_{1,\alpha }\left(x\;|\;A\left(x\right)\right).$ This implies that the weights are comparable as they are all concerned with predicting essentially the same data and moreover they are not concerned with predicting aspects of the data that have no relation to the quantity of interest. In Context II this is not necessarily the case which raises the question of whether or not the weights are comparable.

It is not obvious how to deal with this issue in general, but in some contexts the structure of the models is such that x↔(L(x),A(x)) where *L* has fixed dimension and *A* is ancillary for each model. For example, if all the models are location models, then $x={\left(x_{1},\ldots ,x_{n}\right)}^{\prime }=\bar{x}1_{n}+A\left(x\right),$ where $1_{n}$ is a column of 1’s, and $A\left(x\right)={\left(x_{1}-\bar{x},\ldots ,x_{n}-\bar{x}\right)}^{\prime }$ is ancillary. In such a case, it is desirable to determine the weights based on how well the inference bases predict the value of L(x) and not A(x). To take account of this it is necessary that Jeffrey conditionalization be modified so that the *i*-th posterior weight is now proportional to $\alpha_{i}m_{i}\left(x\;|\;A\left(x\right)\right)$ where $m_{i}\left(\cdot \;|\;A\left(x\right)\right)$ is the *i*-th prior predictive of the data given A(x). Examples 4 and 5 illustrate this modification.

While Proposition 6 does not apply with the conditional weights, a similar result can be proved and for this some assumptions are imposed to simplify the proof. Let the basic sample space be such that there is a finite ancillary partition $(B_{1},\ldots ,B_{m}),$ applicable to each of the *k* models, and for any *n* the ancillary is given by $A\left({\overset{˘}{x}}_{n}\right)=\left(n_{1}\left({\overset{˘}{x}}_{n}\right),\ldots ,n_{m}\left({\overset{˘}{x}}_{n}\right)\right)$ where $n_{i}\left({\overset{˘}{x}}_{n}\right)$ records the number of values in the sample that lie in $B_{i}.$ Then the probability distribution of $A({\overset{˘}{x}}_{n})$ for the *i*-th model is given by the multinomial $\left(n,p_{i1},\ldots ,p_{im}\right)$ where the $p_{ij}$ are fixed and independent of the model parameter. Denote this probability function at the observed data by $f_{i}\left(\overset{˘}{n}\left({\overset{˘}{x}}_{n}\right)\right)$ where $\overset{˘}{n}\left({\overset{˘}{x}}_{n}\right)=\left(n_{1}\left({\overset{˘}{x}}_{n}\right),\ldots ,n_{m}\left({\overset{˘}{x}}_{n}\right)\right).$ Suppose that each parameter space $\Theta_{i}$ is finite with the prior $\pi_{i}$ everywhere positive. Let $\alpha_{i}\propto \alpha_{i}^{*}/f_{i}\left(\overset{˘}{n}\left({\overset{˘}{x}}_{n}\right)\right)$ for $\alpha^{*}\in S_{k}$ and *J* denote the set of indices containing the true distribution. Calling these requirements condition ★, the following is proved in the Appendix A.

**Proposition 7.** *If condition* ★ *holds, then*
$$\alpha_{i}m_{i}\left({\overset{˘}{x}}_{n}\right)/m_{1,\alpha }\left({\overset{˘}{x}}_{n}\right)\to w_{i}=I_{J}\left(i\right)\alpha_{i}^{*}\pi_{i}\left(\theta_{itrue}\right)/\sum_{j\in J}\alpha_{j}^{*}\pi_{j}\left(\theta_{jtrue}\right)\geq 0$$
*and $\sum_{i=1}^{k}w_{i}=1.$*

Proposition 7 provides the desirable consistency result as the only thing that is affected here are the weights which have been shown to have the correct asymptotic property.

Of course, this result needs to be generalized to handle even a situation like the location model. For this some conditions on the models and priors are undoubtedly required but this is not pursued further here. One key component of the proof is the existence of the ancillary partition $\left(B_{1},\ldots ,B_{m}\right)$ and such a structural element seems necessary generally to obtain the comparability of the weights. In group-based models, like linear regression and many others, such a structure exists via the usual ancillaries, see Example 5. As an approximation, a finite ancillary partition can be constructed via the ancillary statistic in question and so Proposition 7 is applicable. It should also be noted that, if the original models are replaced by the conditional models given the ancillary, then (Equation 4) gives the same answer as this modification as the values of $RB_{i,\Psi }\left(\psi \;|\;x\right)$ are unaffected by the conditioning.

Clearly there are connections with the combination rule for statistical evidence advocated here and Bayesian model averaging as discussed in [33]. In fact, the posterior (Equation 5) is the same as that obtained from Bayesian model averaging. The focus here, however, is on the inferences that arise from a direct measure of statistical evidence rather than basing these on the posterior alone and these inferences are different. That posterior probabilities do not provide a suitable measure of evidence can be seen from simple examples such as the Prosecutor’s Fallacy as discussed in [28] (Example 4). It is shown there that the posterior probability of of an event (guilt) being true can be very small but there is still clear evidence that the event is true. So, this is only weak evidence because the posterior probability indicates a small belief in what the evidence indicates. As has been demonstrated here, the consensus preserving feature supports the linear rule over other possible candidates for combining and this, together with Jeffrey conditionalization, also supports the posterior (5) obtained via Bayesian model averaging. Issues concerned with the comparability of the weights remain to be more fully addressed for both methodologies.

## 5. Examples

Some examples are now considered that demonstrate a number of considerations.

**Example 2.** 
*Location-normal model with normal priors.*

*Suppose $x=(x_{1},\ldots ,x_{n})$ is a sample from a $N(\mu ,\sigma_{0}^{2})$ distribution where the mean is unknown but the variance is known. It might be more appropriate to model this with an unknown variance but this situation will suffice for illustrative purposes and there are applications for it in physics, where the variation arising from a given measurement process is well understood. The model is then given by, after reducing to the mss $\bar{x}$, the collection of $N(\mu ,\sigma_{0}^{2}/n)$ distributions and so this is Context I. Suppose there are three analysts and they express their priors for μ as $N(\mu_{i},\tau_{i}^{2})$ distributions for i=1,2,3 sothe i-th posterior is $N({\left(n/\sigma_{0}^{2}+1/\tau_{i}^{2}\right)}^{-1}\left(n\bar{x}/\sigma_{0}^{2}+\mu_{i}/\tau_{i}^{2}\right),{\left(n/\sigma_{0}^{2}+1/\tau_{i}^{2}\right)}^{-1})$ and these ingredients determine the relative belief ratios. For combining, the prior predictives are also needed and the i-th prior predictive density $m_{i}$ for $\bar{x}$ is the $N(\mu_{i},\sigma_{0}^{2}/n+\tau_{i}^{2})$ density. Suppose the inference bases are equally weighted, so the posterior weight of the i-th analysis relative to the others is determined by how well the observed value $\bar{x}$ fits the $N(\mu_{i},\sigma_{0}^{2}/n+\tau_{i}^{2})$ distribution. Note, however, that even if there is a perfect fit, in the sense that $\bar{x}=\mu_{i},$ the weight still depends on the quantity $\sigma_{0}^{2}/n+\tau_{i}^{2}.$ For example, if the $\mu_{i}$ are all equal and there is a perfect fit, then the i-th weight is proportional to ${\left(1+n\tau_{i}^{2}/\sigma_{0}^{2}\right)}^{-1/2}$ and this weight goes to 0 as $\tau_{i}^{2}\to \infty$ with the other prior variances constant and goes to its biggest value when $\tau_{i}^{2}\to 0.$ This suggests that making a prior quite diffuse leads to reducing the impact the corresponding inference base has in the combined analysis.*

*Consider a specific data example where the true value is μ=10, with $\sigma_{0}=1$, and sample sizes n=5,10,25,100. Data were generated from the true distribution obtaining the values $\bar{x}=10.92,9.87,9.96,10.12$ respectively. For the priors, we use $\left(\mu_{1},\tau_{1}^{2}\right)=\left(12,2\right),\left(\mu_{2},\tau_{2}^{2}\right)=\left(9,1\right),\left(\mu_{3},\tau_{3}^{2}\right)=\left(11,4\right)$ equally weighted. Figure 1 plots the combined prior, posterior and relative belief ratio for the n=10 case. Table 1 records the estimates of μ, the plausible regions together with the posterior and prior contents of these intervals for each inference base and linear pooling. Note that, in this case, because the model is the same for each inference base and μ is the model parameter, the estimates are all equal to the MLE of μ but the plausible intervals and their posterior contents differ.*


Consider now prediction which produces the interesting consequence that Context II now obtains even when all the models are same.

**Example 3.** 
*Prediction.*

*Consider Context I but suppose interest is in predicting a future value y∈Y, whose distribution is conditionally independent of the observed data x given θ and has model $\{g_{\lambda }:$λ∈Λ} where Λ:Θ→Λ with $\lambda_{true}=\Lambda \left(\theta_{true}\right).$ The first step in solving this problem is to determine the relevant inference bases and this is carried out by integrating out the nuisance parameter which in this case is θ. So the i-th inference base is given by $I_{i}=\left(x,\left\{m_{i}\left(\cdot \;|\;y\right):y\in Y\right\},m_{i,Y}\right)$ where $m_{i,Y}$ is the density of the i-th prior for y, namely, $m_{i,Y}\left(y\right)=\int_{\Theta }g_{\Lambda (\theta )}\left(y\right)\pi_{i}\left(\theta \right)\;d\theta ,$ and $m_{i}\left(x\;|\;y\right)=\int_{\Theta }f_{\theta }\left(x\right)g_{\Lambda (\theta )}\left(y\right)\pi_{i}\left(\theta \right)\;d\theta /m_{i,Y}\left(y\right)$ is the conditional density of x given y. Note that unconditionally x and y are not independent and now the collection of possible distributions for x is indexed by y. The i-th posterior density of y is then $m_{i,Y}\left(y\;|\;x\right)=m_{i}\left(x\;|\;y\right)m_{i,Y}\left(y\right)/m_{i}\left(x\right).$*

*The models $\left\{m_{i}\left(\cdot \;|\;y\right):y\in Y\right\}$ are now not all the same so this is Context II with common data as discussed in Section 4. It is assumed, as is typically the case, that the mss for these models is constant in i so the weights are comparable. Applying (Equation 4), with the single data set x, leads to*

$$RB_{1,\alpha ,Y}^{*}\left(y\;|\;x\right)=\sum_{i=1}^{k}\frac{\alpha_{i}m_{i}\left(x\right)}{m_{1,\alpha }\left(x\right)}RB_{i,Y}\left(y\;|\;x\right)$$

*with $RB_{i,Y}\left(y\;|\;x\right)=m_{i,Y}\left(y\;|\;x\right)/m_{i,Y}\left(y\right)=m_{i}\left(x\;|\;y\right)/m_{i}\left(x\right)$ and (Equation 5) leads to posterior $m_{1,\alpha ,Y}^{*}\left(y\;|\;x\right)=\sum_{i=1}^{k}\left(\alpha_{i}m_{i}\left(x\right)/m_{1,\alpha }\left(x\right)\right)m_{i,Y}\left(y\;|\;x\right).$ Note that in this case the posterior of y given x is well-defined via Bayesian conditioning and equals $m_{1,\alpha ,Y}^{*}\left(y\;|\;x\right)$ so there is no need to invoke Jeffrey’s conditionalization for the posterior. It is notable, however, that if the relative belief ratio for y is computed using this posterior and the prior $m_{1,\alpha ,Y}\left(y\right)=\sum_{i=1}^{k}\alpha_{i}m_{i,Y}\left(y\right),$ then this equals*

(6)
$$\sum_{i=1}^{k}\frac{\alpha_{i}m_{i}\left(x\right)m_{i,Y}\left(y\right)}{m_{1,\alpha }\left(x\right)m_{1,\alpha ,Y}\left(y\right)}RB_{i,Y}\left(y\;|\;x\right)$$

*which does not equal $RB_{1,\alpha ,Y}^{*}\left(y\;|\;x\right).$ Given that the weights in (Equation 6) depend on the object of interest y, this does not correspond to linear pooling of the evidence and this is because the model is not constant. There is no reason to suppose that (Equation 6) will retain the good properties of linear pooling and experience with it suggests that it is not the correct way to combine. As such, the recommended approach is via (Equation 4) based on Jeffrey’s conditionalization and which retains the good properties of linear pooling.*

*Suppose now the context is as discussed in Example 2 but the goal is to make a prediction concerning a future independent value $y∼N(\mu ,\sigma_{0}^{2}).$ So the i-th prior $m_{i,Y}$ is given by $y∼N(\mu_{i},\sigma_{0}^{2}+\tau_{i}^{2})$ and the i-th posterior $m_{i,Y}\left(\cdot \;|\;\bar{x}\right)$ is $y\;|\;\bar{x}∼N\left({\left(n/\sigma_{0}^{2}+1/\tau_{i}^{2}\right)}^{-1}\left(n\bar{x}/\sigma_{0}^{2}+\mu_{i}/\tau_{i}^{2}\right),{\left(n/\sigma_{0}^{2}+1/\tau_{i}^{2}\right)}^{-1}+\tau_{i}^{2}\right).$ Table 2 gives the results for predicting y using the data in Example 2. The final row indicates what happens as n→∞ and note that the weights converge as well with the i-th limiting weight proportional to ${\left(\sigma_{0}^{2}+\tau_{i}^{2}\right)}^{-1/2}\exp\left(-{\left(\mu -\mu_{i}\right)}^{2}/2\left(\sigma_{0}^{2}+\tau_{i}^{2}\right)\right)$ which depends on the relative accuracy of the i-th prior with respect to the true mean μ. When all the prior variances are the same, the prior which has its mean closest to the true value will give the heaviest weight. Also, as $\tau_{i}^{2}\to \infty$ the i-th weight goes to 0. Note that the limiting plausible intervals are dependent on the prior and the interval does not shrink to a point because y is random. The limiting posterior content of these intervals is the probability content given by the true distribution of y.*

*For the limiting plausible intervals for y to still be dependent on the prior is different than the situation when making inference about a parameter as, in that case, the plausible intervals shrink to the true value as the amount of data increases. The difference is that there is not a “true” value for y. The limiting plausible interval does not allow for all possible values for y and the effect of the prior is to disallow some possible values because belief in such a value is less than that specified by the prior of y. As can be seen from Table 2 this effect is not great unless the prior, as with $\pi_{1}$ here, puts little mass near the true value. However, such an occurrence also reduces the limiting weight for such a component.*


Consider now an example where the weights require adjustment.

**Example 4.** 
*Location-normal models with different variances.*

*Consider a situation similar to Example 2 but now with three distinct models so this is Context II. Here the i-th statistician assumes that the true distribution is $N(\mu ,\sigma_{i0}^{2})$ where the $\sigma_{i0}^{2}$ are known but $\mu \in R^{1}$ is unknown and interest is in ψ=Ψ(μ)=μ. The same three $N(\mu_{i0},\tau_{i0}^{2})$ priors are assumed as in Example 2. So the statisticians disagree about the “known” variance of the sampling distribution and an ancillary needs to play a role to make the weights comparable.*

*In this case $A\left(x\right)=x-\bar{x}1$ is ancillary for each model and is independently distributed from the common mss $L\left(x\right)=\bar{x}∼N\left(\mu ,\sigma_{i0}^{2}/n\right)$ and x↔(L(x),A(x)). Therefore, with equal weights for the priors, and taking the ancillaries into account, the i-th weight satisfies*

$$m_{i}\left(L\left(x\right)\;|\;A\left(x\right)\right)\propto {\left(\sigma_{i0}^{2}/n+\tau_{i}^{2}\right)}^{-1/2}\varphi \left({\left(\sigma_{i0}^{2}/n+\tau_{i}^{2}\right)}^{-1/2}\bar{x}\right).$$

*From this it is seen that the assumed variances and the prior both play a role in determining how much weight a given analysis should have. Note that as $\sigma_{i0}^{2}\to \infty$ or $\tau_{i}^{2}\to \infty$, and all other parameters are fixed, then the weight of the i-th analysis goes to 0 as it should as, in the limit, no information is being provided about the true value of μ. Proposition 6 tell us that when n→∞ and the i-th variance is correct and the others are not, then the i-th inference base will dominate.*


Consider now an example where the models are truly different.

**Example 5.** 
*Location with quite different models.*

*Consider again the context of Example 2 but suppose that one of the models, say the one in $I_{1},$ is a $t_{1}$ (Cauchy) location model, while the other models and all the priors are as previously specified. For all three inference bases $A\left(x\right)=x-\bar{x}1$ is ancillary. To ensure that $\sigma_{0}$ has the same interpretation across all inference bases, the $t_{1}$ density is rescaled by $\eta_{0}$ so that the interval $\left(-\sigma_{0},\sigma_{0}\right)$ contains 0.6827 of the probability for all 3 distributions. This implies $\eta_{0}=\sigma_{0}/\tan\left(0.1827\pi \right)$ and, with $g\left(z\right)=1/\eta_{0}\pi \left(1+z^{2}/\eta_{0}^{2}\right),$ the first model is $\left\{f_{\mu }\left(x\right):\mu \in R^{1}\right\}$ where $f_{\mu }\left(x\right)=\prod_{i=1}^{n}g\left(x_{i}-\mu \right).$ To obtain the corresponding weight, the following expression needs to be evaluated numerically,*

$$m_{1}\left(x\;|\;A\left(x\right)\right)=\frac{\int_{-\infty }^{\infty }f_{1,\mu }\left(\left(\bar{x}-\mu \right)1+A\left(x\right)\right)\pi_{1}\left(\mu \right)\;d\mu }{\int_{-\infty }^{\infty }\int_{-\infty }^{\infty }f_{1,\mu }\left(\left(\bar{x}-\mu \right)1+A\left(x\right)\right)\pi_{1}\left(\mu \right)\;d\mu \;d\bar{x}}.$$

*When applied to the data of Example 2 very similar results are obtained. Table 3 contains the weights for the inference bases for this situation.*


The following example is of considerable practical importance.

**Example 6.** 
*Linear regression.*

*Suppose that the data is $\left(x_{i},y_{i}\right)$ for i=1,…,n and there are two analysts where both propose a simple regression model y=Xβ+σz where $X=(1_{n}/\sqrt{n},x)$ with $1_{n}⊥x,$$\left||x|\right|=1,\beta ={\left(\beta_{1},\beta_{2}\right)}^{\prime }\in R^{2}$ and σ>0 unknown and z is a sample from N(0,1) for analyst 1 and is a sample from a $t_{\lambda }/\sqrt{(\lambda -2)/\lambda }$ distribution for analyst 2 for some value λ>2. In both models $\sigma^{2}$ is the variance of a $y_{i}.$ Letting $b={\left(X^{\prime }X\right)}^{-1}X^{\prime }y$ be the least squares estimate of β and $s^{2}=||y-Xb||,$ then y↔(L(y),A(y)) where $L\left(y\right)=\left(b,s^{2}\right)$ and A(y)=(y−Xb)/s is ancillary for both models. Further suppose that the quantity of inferential interest is the slope parameter $\psi =\Psi \left(\beta_{1},\beta_{2},\sigma^{2}\right)=\beta_{2}.$ Denoting the relevant density of a $z_{i}$ by f, the joint density of (b,s) given A(y)=a is proportional to*

$$s^{n-3}\sigma^{-n}\prod_{i=1}^{n}f\left(\frac{b_{1}-\beta_{1}}{\sigma }+\frac{b_{2}-\beta_{2}}{\sigma }x_{i}+\frac{s}{\sigma }a_{i}\right).$$

*The posterior density of $\beta_{2}$ can be worked out in closed-form when f is the N(0,1) density but generally it will require numerical integration to determine the posterior density and the posterior weights for the combination.*

*For the prior, suppose both analysts agree on $\beta \;|\;\sigma^{2}∼N_{2}\left(0,\tau_{0}^{2}\sigma^{2}I\right)$ and $1/\sigma^{2}∼$ gamma_rate_
$\left(\alpha_{1},\alpha_{2}\right).$ Note that the zero mean for β may entail subtracting a known, fixed constant vector from y so this, and the assumption that $1_{n}⊥x,$ may entail some preprocessing of the data. The prior distribution of the quantity of interest is then $\beta_{2}∼\tau_{0}\sqrt{\alpha_{2}/\alpha_{1}}t_{2\alpha_{1}}$ where $t_{2\alpha_{1}}$ denotes the t distribution on $2\alpha_{1}$ degrees of freedom.*

*Obtaining the hyperparameters of the prior requires elicitation and this can be carried out using the following method as described in [34]. Suppose that it is known with virtual certainty, based on our knowledge of the measurements being taken, that $\beta_{1}+\beta_{2}x$ will lie in the interval $\left(-m_{0},m_{0}\right)$ for some $m_{0}>0$ for all x∈R a compact set centered at 0 and contained in ${\left[-1,1\right]}^{k}$ on account of the standardization. The phrase ‘virtual certainty’ is interpreted here as a probability greater than or equal to γ where γ is some large probability like 0.99. Therefore, the prior on β must satisfy $2\Phi (m_{0}/\sigma \tau_{0}{\left\{1+x^{2}\right\}}^{1/2})-1\geq \gamma$ for all x∈R which implies*

(7)
$$\sigma \leq m_{0}/\zeta_{0}\tau_{0}z_{(1+\gamma )/2}$$

*where $\zeta_{0}^{2}=1+\max_{x\in R}x^{2}\leq 2$ with equality when R=[−1,1]. An interval that will contain a response value y with virtual certainty, given predictor value x, is $\beta_{1}+\beta_{2}x\pm \sigma z_{(1+\gamma )/2}.$ Suppose that we have lower and upper bounds $s_{1}$ and $s_{2}$ on the half-length of this interval so that $s_{1}\leq \sigma z_{(1+\gamma )/2}\leq s_{2}$ or, equivalently,*

(8)
$$s_{1}/z_{(1+\gamma )/2}\leq \sigma \leq s_{2}/z_{(1+\gamma )/2}$$

*holds with virtual certainty. Combining (Equation 8) with (Equation 7) implies $\tau_{0}=m_{0}/s_{2}\zeta_{0}.$ To obtain the relevant values of $\alpha_{1}$ and $\alpha_{2},$ let $G\left(\alpha_{1},\alpha_{2},\cdot \right)$ denote the cdf of the gamma_rate_
$\left(\alpha_{1},\alpha_{2}\right)$ distribution and note that $G\left(\alpha_{1},\alpha_{2},w\right)=G\left(\alpha_{1},1,\alpha_{2}w\right).$ Therefore, the interval for $1/\sigma^{2}$ implied by (Equation 8) contains $1/\sigma^{2}$ with virtual certainty, when $\alpha_{1},\alpha_{2}$ satisfy $G^{-1}\left(\alpha_{1},\alpha_{2},\left(1+\gamma \right)/2\right)=s_{1}^{-2}z_{(1+\gamma )/2}^{2},G^{-1}\left(\alpha_{1},\alpha_{2},\left(1-\gamma \right)/2\right)=s_{2}^{-2}z_{(1-\gamma )/2}^{2},$ or equivalently*

(9)
$$\begin{array}{cc} G(\alpha_{1},1,\alpha_{2}s_{1}^{-2}z_{(1+\gamma )/2}^{2}) & =(1+\gamma )/2, \end{array}$$


(10)
$$\begin{array}{cc} G(\alpha_{1},1,\alpha_{2}s_{2}^{-2}z_{(1-\gamma )/2}^{2}) & =(1-\gamma )/2. \end{array}$$

*It is a simple matter to solve these equations for $\left(\alpha_{1},\alpha_{2}\right).$ For this choose an initial value for $\alpha_{1}$ and, using (Equation 9), find w such that $G\left(\alpha_{1},1,w\right)=\left(1+\gamma \right)/2,$ which implies $\alpha_{2}=ws_{1}^{2}/z_{(1+\gamma )/2}^{2}.$ If the left-side of (Equation 10) is less (greater) than (1−γ)/2, then decrease (increase) the value of $\alpha_{1}$ and repeat step 1. Continue iterating this process until satisfactory convergence is attained.*

*Consider now a numerical example drawn from [35] where the response variable is income in U.S. dollars per capita (deflated), and the predictor variable is investment in dollars per capita (deflated) for the United States for the years 1922–1941. The data are provided in Table 4. The data vector y was replaced by $y-X{\left(340,3\right)}^{t}$ as this centered the observations about 0. Taking $\gamma =0.99,\zeta_{0}=\sqrt{2},m_{0}=30,s_{1}=10,s_{2}=40$ leads to the values $\tau_{0}=0.54,\alpha_{1}=4.05,\alpha_{2}=140.39.$ The following prior is then used for both models,*

$$\left(\beta_{1},\beta_{2}\right)\;|\;,\sigma^{2}∼N_{2}\left(0,{\left(0.54\right)}^{2}\sigma^{2}I\right),\;1/\sigma^{2}∼\mathrm{gamma}\left(4.05,140.39\right).$$

*Table 5 presents the weights that result when different $t_{\lambda }$ error distributions are considered to be combined with the results from a N(0,1) error assumption. Presumably this arises when one analyst is concerned that tails longer than the normal are appropriate. As can be seen, the normal error assumption dominates except for λ=100 when the inferences do not differ by much in any case. This is not surprising as various residual plots do not indicate any issue with the normality assumption for these data. These weights were computed using importance sampling and were found to be robust to the prior by repeating the computations after making small changes to the hyperparameters.*

*The approach taken in this example is easily generalized to more general linear regression models including situations where the priors change.*


## 6. Conclusions

The problem of how to combine evidence has been considered for a Bayesian context where each analyst proposes a model and prior for the same data. Linear opinion pooling is seen as the natural way to make such a combination, at least when the inference bases only differ in the priors on the parameter of interest. This has been shown to have appropriate properties such as preserving a consensus with respect to the evidence and, when combining evidence is considered as opposed to just combining priors, behaves appropriately when considering independent events. In certain contexts the idea can be extended in a logical way based on the idea underlying Jeffrey conditionalization. This approach has been shown to behave correctly asymptotically in a wide variety of situations.

There are a number of factors that need to be considered when implementing the methods discussed here. As mentioned in the Introduction, we have assumed that each of the sampling models and priors used have been subjected to model checks and checking for prior–data conflict, respectively. As such, we are not considering combining the evidence obtained from contexts where the ingredients are contradicted by the data, and this is to be regarded as a key part of the analysis. It is also worth noting too that [36] establishes that relative belief inferences are optimally robust to the choice of the prior on Ψ, and so, provided there is no prior–data conflict, a degree of robustness to the used priors can be expected. This does not, however, address issues concerned with sensitivity to the sampling models or to the priors used for nuisance parameters. There are also issues that arise for the choice of α. Unless there are good reasons to do otherwise, using uniform weights seems like the best choice as then only the data determines the relative weighting. For Context II, however, as discussed in Section 4, there are general concerns with the comparability of the model weights and that has been only partially addressed here.

The developments here do not cover contexts where there are different data sets and different models. If the models are all for the same basic responses, then one possibility is to simply combine data sets and proceed, as we have demonstrated here. More generally, it may be that the only aspect in common among the models is the characteristic of interest Ψ, and then it is not clear how we should combine this. The combination rule given by$$RB_{1,\alpha ,\Psi }^{**}\left(\psi \;|\;x_{1},\ldots ,x_{k}\right)=\sum_{i=1}^{k}\frac{\alpha_{i}m_{i}\left(x_{i}\right)}{m_{1,\alpha }\left(x_{1},\ldots ,x_{k}\right)}RB_{i,\Psi }\left(\psi \;|\;x_{i}\right),$$
where $m_{1,\alpha }\left(x_{1},\ldots ,x_{k}\right)=\sum_{i=1}^{k}\alpha_{i}m_{i}\left(x_{i}\right),$ suggests itself as a generalization of what has been considered here. Further investigation is required, however, as it is necessary to ensure that the weights $\alpha_{i}m_{i}\left(x_{i}\right)/m_{1,\alpha }\left(x_{1},\ldots ,x_{k}\right)$ are indeed comparable, so some modification is probably required that is context-dependent.

It does not seem essential that we restrict attention to combination rules based on power mean priors as we have done here. For example, one could consider combining the $RB_{i,\Psi }\left(\psi \;|\;x_{i}\right)$ themselves according to some rule. For example, a power rule could be used to combine these quantities. Even in Context I, however, this loses the interpretation of the combination as a valid measure of evidence through the principle of evidence. Of course, (Equation 3) arises in both such approaches to the problem and probably should, no matter which generalized rule is adopted, when it is applied to Context I.

The problem of combining evidence is an important one in science, as evidenced by extensive discussion in the literature over many years. What has been shown here is that a very natural definition of how to measure statistical evidence can lead to a natural solution in a number of significant contexts.

## Figures and Tables

**Figure 1 entropy-27-00654-f001:**
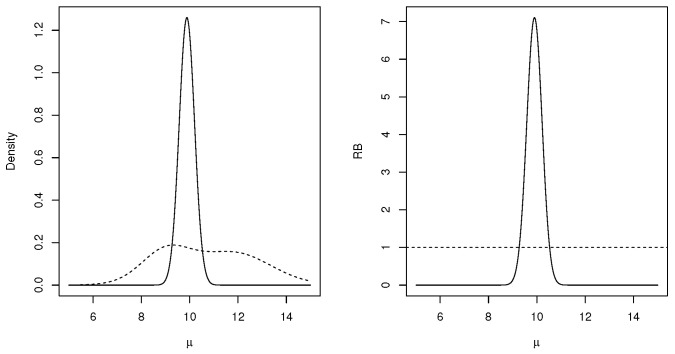
Plots of prior (- - -) and posterior (—) densities in the left panel and, in the right panel, plots of the relative belief ratio (—) for μ and constant 1 (- - -) in Example 2 when n=10.

**Table 1 entropy-27-00654-t001:** Relative belief estimates, plausible intervals (posterior weights, and contents underneath) for μ in Example 2.

*n*	Estimate	$I_{1}$	$I_{2}$	$I_{3}$	Combination
5	10.9	(10.2,11.7) 0.431,0.92	(9.9,11.9) 0.164,0.95	(10.1,11.7) 0.406,0.93	(10.0,11.7) 0.93
10	9.9	(9.2,10.6) 0.176,0.98	(9.3,10.4) 0.507,0.93	(9.2,10.5) 0.317,0.96	(9.3,10.5) 0.95
25	10.0	(9.5,10.4) 0.192,0.98	(9.6,10.4) 0.478,0.96	(9.5,10.4) 0.330,0.97	(9.5,10.4) 0.97
100	10.1	(9.9,10.4) 0.229,0.99	(9.9,10.4) 0.418,0.99	(9.9,10.4) 0.354,0.99	(9.9,10.4) 0.99

**Table 2 entropy-27-00654-t002:** Relative belief estimates, plausible intervals (posterior contents underneath) for *y* in Example 3 with the posterior weights as in Table 1.

*n*	Prediction	$I_{1}$	$I_{2}$	$I_{3}$	Combination
5	10.9	(8.7,12.1) 0.82	(9.7,16.0) 0.81	(9.4,12.4) 0.83	(9.2,13.7) 0.94
10	9.9	(6.5,11.1) 0.87	(9.03,12.5) 0.79	(8.0,11.2) 0.86	(7.7,11.8) 0.94
25	10.0	(6.7,11.2) 0.87	(9.1,12.7) 0.79	(8.2,11.3) 0.86	(7.9,11.9) 0.95
100	10.1	(7.1,11.3) 0.87	(9.3,13.2) 0.80	(8.4,11.4) 0.86	(8.1,12.2) 0.96
*∞*	10.0	(6.9,11.2) 0.88	(9.2,12.8) 0.79	(8.2,11.3) 0.87	(8.0,12.0) 0.96

**Table 3 entropy-27-00654-t003:** Weights for the inference bases in Example 5.

*n*	$I_{1}$	$I_{2}$	$I_{3}$
5	0.4248	0.1652	0.4100
10	0.2036	0.4900	0.3064
25	0.1716	0.4898	0.3386
100	0.2235	0.4202	0.3563

**Table 4 entropy-27-00654-t004:** Haavelmo’s data on income and investment from Zellner (1996) used in Example 6.

Year	Income	Investment	Year	Income	Investment
1922	433	39	1932	372	22
1923	483	60	1933	381	17
1924	479	42	1934	419	27
1925	486	52	1935	449	33
1926	494	47	1936	511	48
1927	498	51	1937	520	51
1928	511	45	1938	477	33
1929	534	60	1939	517	46
1930	478	39	1940	548	54
1931	440	41	1941	629	100

**Table 5 entropy-27-00654-t005:** Weights for normal and $t_{\lambda }$ errors in Example 6.

λ	100	50	20	10	5	3
N(0,1)	0.556	0.612	0.766	0.928	0.998	1.000
$t_{\lambda }$	0.444	0.388	0.234	0.072	0.002	0.000

## Data Availability

All data used is available within the paper.

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
