# Peer review of "Combining Statistical Evidence When Evidence Is Measured by Relative Belief"

_entropy, 2025, doi:10.3390/e27060654_

Round 1
Reviewer 1 Report
Comments and Suggestions for Authors
An excellent piece of work.
Author Response
Thank-you for the endorsement.
Reviewer 2 Report
Comments and Suggestions for Authors
To be honest, I am not sure whether I am entitled conduct a review on such paper. It covers discussions on some philosophical concepts in Bayesian Statistics, in which I had no serious thoughts and reflections. (I could have turned down the review request already, but I do not know how to do this)
So please take my review as an auxilary one since I could not give any comment other than editorial issues.
line 193: thereafter -> thereafter.
Author Response
Thanks, problem fixed.
Reviewer 3 Report
Comments and Suggestions for Authors
The manuscript “Combining statistical evidence when evidence is measured by relative belief” threats the combining statistical evidence, contained in multiple Bayesian inference bases, given by same data and model, different priors in Context I and generalized case in Context II, where data and model are also varying. Author addresses the question of consensus preserving rule for combining statistic evidence in Context I discuss the selection of the prior weighting for combining statistic evidence. Furthermore, also the generalization to the Context II and occurring difficulties are discussed in the manuscript. The theoretical findings are illustrated by six application examples.
The manuscript is well written and provides substantial scientific contribution. However, I have some minor remarks to the manuscript, in particular, to the notation used.
Remarks
- P.2, first equation: Here and later in the manuscript (conditional) prior is denoted by $\Pi$ and / instead of by $\pi$.
- L.164/190: I have missed the definition of $\Pi_i$, should it probably be the $\pi_i$?
- Proposition 3: The explanation of the relationship between $RB_i$, and $RB_t,\alpha$ would be, in my opinion, helpful for the reader. Now, I am missing also the definitions of them.
- 327: $\Pi_i$ or $\pi_i$?
- Eq. (3): I’m missing the definition of $RB_i,\Psi$.
Author Response
- P.2, first equation: Here and later in the manuscript (conditional) prior is denoted by $\Pi$ and / instead of by $\pi$. L.164/190: I have missed the definition of $\Pi_i$, should it probably be the $\pi_i$?
This has been fixed in the first equation. Also, I have added the following paragraph to the end of the Introduction.
"Throughout the paper the densities of probability distributions will be represented by lower case symbols and the associated probability measure will be represented by the same symbol in upper case. For example, if a prior density is denoted by $\pi$, then the prior probability measure will be denoted by $\Pi$ with the posterior density denoted $\pi(\cdot\, | \, x)$ and the posterior probability measure by $\Pi(\cdot\, | \, x)$."
- Proposition 3: The explanation of the relationship between $RB_i$, and $RB_t,\alpha$ would be, in my opinion, helpful for the reader. Now, I am missing also the definitions of them.
I have now defined $RB_{\Psi}(\psi\,|\,x)$ in a displayed equation just after the principle of evidence. Also, I have now explicitly defined these quantities in the following paragraph which appears just before Proposition 3.
"The next result examines the behavior of the combination rules of Section 2 with respect to evidence and is stated initially for the full model parameter $\theta$ in Context I. For this $RB_{i}(\theta\,|\,x)$ is the relative belief ratio for $\theta$ that results from the $i$-th inference base $\mathcal{I}_{i}=(x_{i},\mathcal{M}_{i},\pi_{i})$ and $RB_{t,\alpha}(\theta\,|\,x)$ is the relative belief ratio for $\theta$ that results from combining the $k$ priors using the $t$-th power mean combination rule."
The prior predictive densities $m_{1,\alpha}(x)$ and $m_{t,\alpha}(x)}$ have been defined previously in the paper.
- 327: $\Pi_i$ or $\pi_i$?
The distinction in these symbols has now been clarified in the final paragraph of the Introduction so $\Pi_i$ is the correct symbol as it denotes the posterior probability measure. - Eq. (3): I’m missing the definition of $RB_i,\Psi$. I have now defined this quantity after Eq. (3) as in "where $RB_{i,\Psi}(\psi\,|\,x)$ is the relative belief ratio for $\psi$ obtained from the $i$-th inference base."
Thank-you for your comments.
Reviewer 4 Report
Comments and Suggestions for Authors
The paper presents a well-structured approach to combining statistical evidence from multiple Bayesian inference bases using the relative belief ratio as a measure of evidence. The paper is clearly written and well-motivated, and the examples illustrate the main ideas effectively. I only have three suggestions.
(i) It would be great if the discussion on practical implications (or a real-data application) and potential limitations (e.g., sensitivity to prior weighting) could be presented.
(ii) The treatment of Context II, while interesting, could benefit from more simulation-based validation.
(iii) The literature review on combining priors is strong, but it would be great if more explicit comparisons with related Bayesian model averaging approaches are discussed. In what ways is the current method stronger or weaker in handling model uncertainty?
Author Response
See Attached report. Thank-you for your comments.
